# Single and Co-Cultures of Proteolytic Lactic Acid Bacteria in the Manufacture of Fermented Milk with High ACE Inhibitory and Antioxidant Activities

Shahram Loghman [1], Ali Moayedi [1,*], Mandana Mahmoudi [1], Morteza Khomeiri [1], Laura G. Gómez-Mascaraque [2] and Farhad Garavand [2,*]

[1] Department of Food Science and Technology, Gorgan University of Agricultural Sciences and Natural Resources, Gorgan 4913815739, Iran

[2] Department of Food Chemistry and Technology, Teagasc Moorepark Food Research Centre, Fermoy, Co., P61 C996 Cork, Ireland

* Correspondence: amoayedi@gau.ac.ir (A.M.); farhad.garavand@teagasc.ie (F.G.)

**Abstract:** In this study, single and co-cultures of proteolytic *Lactobacillus delberueckii* subsp. *bulgaricus* ORT2, *Limosilactobacillus reuteri* SRM2 and *Lactococcus lactis* subsp. *lactis* BRM3 isolated from different raw milk samples were applied as starter cultures to manufacture functional fermented milks. Peptide extracts from fermented milk samples were evaluated after fermentation and 7 days of cold storage for proteolytic, angiotensin-converting enzyme (ACE) inhibitory and antioxidant activity by different methods including 2, 2′-diphenyl-1-picrylhydrazyl (DPPH), ferric-reducing antioxidant power (FRAP), OH-radical scavenging, and total antioxidant (molybdate-reducing activity). The highest proteolysis was found in milk fermented by co-cultures of three strains. Fermentation with the mentioned bacteria increased ACE inhibitory and antioxidant activity of the final products which were dependent on peptide concentration. The crude peptide extract obtained from fermented milk with triple co-culture showed the highest ACE inhibitory activity ($IC_{50}$ = 0.61 mg/mL) which was reduced after 7 days of cold storage ($IC_{50}$ = 0.78 mg/mL). Similar concentration-dependent activities were found in antioxidant activity at different antioxidant assays. Overall, high proteolytic activity resulted in increased ACE inhibitory and antioxidant activities, but the highest activity was not necessarily found for the samples with the highest proteolytic activity. The results of this study suggest the potential of using co-cultures of *L. delberueckii* subsp. *bulgaricus*, *L. reuteri* and *L. lactis* subsp. *Lactis* to manufacture antihypertensive fermented milk.

**Keywords:** antihypertensive activity; bioactive peptide; fermentation; lactic acid bacteria; proteolysis

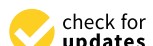



## 1. Introduction

Hypertension (high blood pressure) is considered one of the major problems in human health that increase the risk of cardiovascular diseases, stroke, and sometimes death [1]. Angiotensin-converting enzyme (ACE) converts angiotensin I to angiotensin II, which increases sodium retention and blood pressure. ACE inhibitors decrease the generation of angiotensin II, and therefore prevent hypertension. Synthetic ACE inhibitor drugs are suggested to treat hypertension-related diseases, but these can trigger side effects such as dry cough, hyperkalemia, and angioedema [2]. Therefore, controlling blood pressure through the use of natural compounds such as bioactive peptides is recommended to avoid the side effects of synthetic antihypertensive drugs [3].

Different studies show an association between oxidative stresses and several diseases such as cancer, cardiovascular, neurological, respiratory, kidney, and hypertension diseases [4,5]. Antioxidants prevent the damage resulting from oxidative stress by inhibiting the activity of free radicals [6]. Nowadays, natural antioxidants are preferred by consumers over synthetic antioxidants due to their lack of side effects. Previous studies have reported

that the consumption of natural antioxidants was effective in the improvement of hypertension and cardiovascular diseases in humans [7,8]. Accordingly, new therapies based on the use of natural antioxidants are a new insight into the treatment of hypertension [5]. Recently, ACE inhibitory and antioxidant peptides have been widely studied and suggested as replacements for synthetic antioxidant and antihypertensive compounds.

Proteolytic lactic acid bacteria (LAB) have a unique place among the starter or adjunct cultures used in the dairy and fermentation industries [9]. Their proteolytic system consists of cell envelope proteinase (CEP), peptide transporters, and different peptidases releasing amino acids and peptides from milk proteins [10]. Proteolytic LAB strains reduce milk and dairy allergenicity by breaking down casein during fermentation [11]. Moreover, they can improve the digestibility of protein in fermented dairy products through their proteolytic system [10]. During recent decades, proteolytic LAB strains have gained increasing attention as they have the capability to release bioactive peptides with different physiological activities depending on their structure and amino acid sequences. Some of these bioactive peptides produced by proteolytic LAB during milk fermentation have shown anticancer, antihypertensive, antioxidant, and antidiabetic activities [12–15]. On the other hand, some researchers have used co-cultures of LAB for the fermentation process to assess their synergistic effects [16–18]. They reported that the co-culture of LAB not only improved the fermentation ability of strains without negative effects on the final quality of the product but also increased the production of bioactive peptides. Therefore, finding new strains with technological and functional properties is of interest for the dairy industry. Moreover, comparing the activities of single cultures with those of co-cultures would be helpful for selecting the optimal combination for the production of bioactive compounds with health benefits in fermented milk products.

In this study, proteolytic LAB strains including *Lactobacillus delbrueckii* subsp. *bulgaricus* ORT2, *Limosilactobacillus reuteri* SRM2, and *Lactococcus lactis* subsp. *lactis* BRM3 previously isolated from cow, ewe, and goat raw milk were used as single or co-cultures to manufacture functional fermented milk. ACE inhibitory and antioxidant activities of crude peptide extracts obtained from fermented milk samples were evaluated. The effect of 7-day cold storage on the mentioned functional properties was also examined.

## 2. Materials and Methods

### 2.1. Chemicals and Bacterial Strains

2, 2′-Diphenyl-1-picrylhydrazyl (DPPH), Orthophetalaldehyde (OPA), Hyppuril-L-histidyl-L-leucine (HHL), and ACE enzyme (from rabbit lung) were purchased from Sigma (St. Louis, MO, USA). L-serine was obtained from Bio Basic Inc. (Markham, ON, Canada). Dithioteritol (DTT), tryptone, microbial culture media, and other chemicals were purchased from Merck (Darmstadt, Hesse, Germany). *L. delbrueckii* subsp. *bulgaricus* ORT2 (isolated from goat milk), *L. reuteri* SRM2 (ewe milk), and *L. lactis* subsp. *lactis* BRM3 (cow milk) with accession numbers ON746657, ON746661, and ON746656, respectively, had been previously identified as proteolytic LAB, and kept in the microbial collection (Department of Food Science and Technology, Gorgan University of Agricultural Sciences and Natural Resources, Gorgan, Iran).

### 2.2. Preparation of Fermented Milk Samples and Crude Peptide Extracts

Reconstituted skim milk (12%, *w/v*) (skim milk was provided from Golestan Pegah factory, Gorgan, Iran) was sterilized at 115 °C for 15 min and cooled down to 37 °C for starter culture inoculation. For the fermentation process, sterile milk was inoculated with 2% (*v/v*) of the suspension (in 0.9% sterilized saline) of each isolate or 1:1 combination of isolates ($10^8$ CFU/mL) and incubated (BD115, Binder, Germany) at 37 °C until the pH reached 4.6. Then, the fermented milk was refrigerated at 4 °C, and stored for 7 days. In this study, seven types of fermented milk were manufactured: milk fermented with *L. delbrueckii* subsp. *bulgaricus* ORT2 (T1), *L. reuteri* SRM2 (T2), *L. lactis* subsp. *lactis* BRM3 (T3), co-cultures of *L. delbrueckii* subsp. *bulgaricus* ORT2 and *L. reuteri* SRM2 (T4), co-cultures

of *L. delbrueckii* subsp. *bulgaricus* ORT2 and *L. lactis* subsp. *lactis* BRM3 (T5), co-cultures of *L. reuteri* SRM2 and *L. lactis* subsp. *lactis* BRM3 (T6), and co-cultures of *L. delbrueckii* subsp. *bulgaricus* ORT2, *L. reuteri* SRM2 and *L. lactis* subsp. *lactis* BRM3 (T7). To consider the effect of acidity on the bioactivities, acidified milk (T8) was prepared as a control by adding 0.1 N lactic acid into sterile reconstituted skim milk until the pH reached 4.6.

To prepare crude peptide extracts, the fermented milk samples were centrifuged at 11,200× *g* for 15 min at 4 °C (model Combi 514R, Hanil Science Industrial, Gimpo-si, South Korea), and the supernatant was passed through a Glass Fiber filter with a 2.0 μm pore-size (Merck) to remove the large particles/aggregates. Then it was sterilized using a 0.45 μm syringe filter (Jet Bio-Filtration, Guangzhou, China). The sterile supernatants were dried in a freeze dryer (Model FDB-5503, Operon, Gimpo-si, Korea) and stored at −20 °C until use [19,20].

### 2.3. Determination of Proteolysis and Peptide Content

Proteolysis was monitored spectrophotometrically (model T80, PG Instruments Ltd., Wibtoft, UK) according to Elfahri et al. [12] using OPA as a reagent. Tryptone was used for the preparation of the standard curve [21] and the results were expressed as mg tryptone equivalent per mL (mg TE/mL).

### 2.4. Determination of ACE Inhibitory Activity

ACE inhibitory activity was measured following a method adapted from Donkor et al. [22], with some modifications. All the solutions including crude peptide extracts, HHL substrate and ACE were prepared in 100 mM borate buffer (pH 8.3) containing 0.3 M NaCl. For the assay, 100 μL of different concentrations (20–80 mg/mL) of peptide extract was mixed well with 120 μL of HHL substrate (5 mM). The reaction was started by adding 20 μL of ACE solution (0.1 U/mL) followed by incubation at 37 °C for 30 min. The reaction was terminated by adding 180 μL of 0.1 M HCl. After that, 1 mL of ethyl acetate was added to the mixture and kept at room temperature for 10 min, before centrifuging it at 4500× *g* for 15 min at 25 °C (Model k2042, centurion scientific, Stoughton, UK). Then, the top layer (ethyl acetate containing hippuric acid) was transferred to a new tube and incubated at 50 °C until ethyl acetate was completely evaporated. Finally, the remaining hippuric acid was dissolved in 600 μL of distilled water, and the absorbance of the solution was read at 228 nm (PG Instruments Ltd., Wibtoft, UK). Distilled water was used instead of peptide mixture to prepare the negative control. The measurements were conducted in triplicate (n = 3), and ACE inhibitory activity was calculated according to the following equation:

$$\text{ACEI Activity} = 1 - \left( \frac{C - D}{A - B} \right) * 100 \tag{1}$$

where C, D, A, and B are the absorbance values for the sample, blank (ACE + peptide), negative control, and peptide mixtures (with no added enzyme), respectively. Finally, ACE inhibitory activity was expressed as $IC_{50}$ value which is defined as the amount of peptide extract (mg/mL, as tryptone equivalent) required to inhibit 50% of ACE activity. A lower $IC_{50}$ values indicates higher ACE inhibitory activity.

### 2.5. Determination of Antioxidant Activities
#### 2.5.1. DPPH Scavenging Activity

The DPPH scavenging activity of peptide extracts was determined according to Jemil et al. [23]. Briefly, 190 μL of the sample (peptide extracts, 20–80 mg/mL), 220 μL of ethanol, and 95 μL of ethanolic DPPH solution (2 mM) were mixed and allowed to stand for 60 min in the dark. Then, the absorbance was measured at 517 nm (PG Instruments Ltd., Wibtoft, UK) against a negative control containing distilled water instead of the peptide.

The percentage of DPPH scavenging activity was calculated according to the following equation:

$$\%\text{DPPH scavenging activity} = \left(1 - \frac{As}{Ac}\right) \times 100 \tag{2}$$

where As is the absorbance of the sample and Ac is the absorbance of the negative control.

### 2.5.2. Ferric-Reducing Antioxidant Power (FRAP)

The reducing power (on Fe (III)) was assayed as described by Yildirim et al. [24]. Briefly, 100 μL of peptide sample, 250 μL potassium phosphate buffer (0.2 M, pH 6.6), and 250 μL of 1% ($w/v$) potassium ferricyanide were mixed. The mixture was incubated for 30 min at 50 °C. Then, 250 μL of 10% trichloroacetic acid (TCA) was added, mixed, and centrifuged at $16,000 \times g$ for 10 min at 25 °C. After that, 250 μL of the supernatant was mixed with 250 μL of distilled water and 50 μL of 0.1% ($w/v$) ferric chloride. After keeping the mixture at room temperature for 10 min, the absorbance was measured at 760 nm (PG Instruments Ltd., Wibtoft, UK).

### 2.5.3. OH-Radical Scavenging Activity

OH-radical scavenging activity was determined according to the method described by Wang et al. [25]. A 200 μL amount of crude peptide extract at different concentrations (20–80 mg/mL) was mixed well with 100 μL of Phenantroline (1.865 mM), 100 μL ferrous sulfate (1.865 mM), and 100 μL EDTA (5 mM). Then, 100 μL $H_2O_2$ (0.1%) was added to the mixture, and incubated at 37 °C for 60 min. Finally, the absorbance was read at 536 nm (PG Instruments Ltd., Wibtoft, UK) against a $H_2O_2$-free solution as the blank. For the preparation of negative control, distilled water was used instead of the peptide extract. OH-radical scavenging activity was calculated as:

$$\text{The OH} - \text{radical scavenging activity} = \frac{(As - Ac)}{(Ab - Ac)} \times 100 \tag{3}$$

where *As*, *Ac*, and *Ab* are the absorbance values for sample (peptide extract), negative control (distilled water instead of peptide extract), and blank (distilled water instead of both peptide and $H_2O_2$), respectively.

### 2.5.4. Total Antioxidant Capacity (Ammonium phosphomolybdate Assay)

The capacity of peptide extracts to reduce $Mo^{+6}$ to $Mo^{+5}$ was evaluated according to the method described by Meshginfar et al. [26]. A 100 μL amount of different concentrations (20–80 mg/mL) of peptide extract was added to 80 μL of phosphomolybdate reagent (28 mM sodium phosphate, 0.6 M sulfuric acid, and 4 mM ammonium molybdate), and incubated at 95 °C for 90 min in a water bath. After cooling to room temperature, the absorbance was read at 695 nm (PG Instruments Ltd., Wibtoft, UK). Distilled water was used instead of peptide extract for the negative control.

### 2.6. Statistical Analysis

Data were analyzed using statistical analysis system (SAS software, v 9.1, SAS Institute Inc., Cary, NC, USA) in a completely randomized design (CRD). The Duncan multiple range test was used to compare the mean values ($p \leq 0.05$).

### 3. Results and Discussion

### 3.1. Proteolysis and Peptide Content in Crude Peptide Extract

The results of proteolysis and peptide content determination in the crude peptide extracts (as mg TE/mL) are shown in Figure 1. At the end of fermentation (day 0), the greatest extent of proteolysis was found in samples T3 and T7 ($6.5 \pm 0.6$ and $6.6 \pm 0.4$ mg TE/mL, respectively), while the lowest proteolytic activity was observed in sample T2 ($2.9 \pm 0.2$ mg TE/mL). Among single cultures, *L. lactis* subsp. *lactis* BRM3 showed signifi-

cantly higher proteolytic activity during fermentation than *L. delbruekii* subsp. *bulgaricus* ORT2 and *L. reuteri* SRM2 (not significantly different to the acidified control, T8). In a previous study, the proteolytic activity of different strains of *L. lactis* subsp. *lactis* during fermentation was remarkable compared to many other LABs [27]. Nielsen et al. [28] in a study on *L. lactis*, *L. helveticus*, *L. acidophilus*, and *S. thermophilus* in fermented milk showed that *L. lactis* had the highest proteolytic activity, which was also found to produce the highest amount of peptides with physiological properties. After 7 days of storage, the peptide content in samples T1 and T5 significantly increased and in samples T3, T6, and T7 significantly decreased ($p < 0.05$). Decreased levels of peptides and amino acids after 7 days of storage may be due to their consumption by living cells for cell growth. In the study of Shori et al. [29] on different samples of yogurt, proteolytic activity did not change after 7 days of cold storage, but after 14 days, a decrease in proteolytic activity was reported.

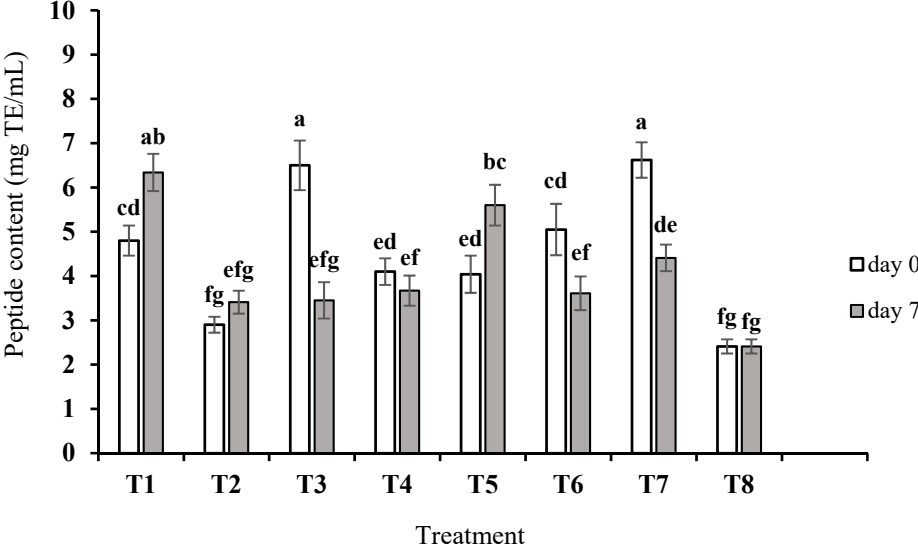

**Figure 1.** Proteolysis extent based on peptide content equivalent to tryptone in fermented milk samples after fermentation (day 0) and after 7 days of storage. T1, T2, and T3 indicate fermented milk with *L. delbruekii* subsp. *bulgaricus* ORT2, *L. reuteri* SRM2, and *L. lactis* subsp. *lactis* BRM3, respectively; T4: fermented milk with *L. delbruekii* subsp. *bulgaricus* ORT2 and *L. reuteri* SRM2; T5: fermented milk with *L. delbruekii* subsp. *bulgaricus* ORT2 and *L. lactis* subsp. *lactis* BRM3; T6: fermented milk with *L. lactis* subsp. *lactis* BRM3 and *L. reuteri* SRM2; T7: fermented milk with *L. delbruekii* subsp. *bulgaricus* ORT2, *L. reuteri* SRM2 and *L. lactis* subsp. *lactis* BRM3; T8: acidified milk with lactic acid. Different letters show significant differences ($p < 0.05$) among all the fermented samples during 7 days of storage.

### 3.2. ACE Inhibitory Activity of Crude Peptide Extract

ACE inhibitory activity of crude peptide extracts at different concentrations (20, 40, and 80 mg/mL) was determined at the end of fermentation and after 7 days of cold storage (Table 1). The intensity of ACE inhibition in crude peptide extracts obtained from single and co-cultures was time- and concentration-dependent and significantly ($p < 0.05$) different. At the end of fermentation, among the samples of milk fermented by single cultures, samples T2 and T3 showed the highest and lowest IC$_{50}$ values ($2.05 \pm 0.15$ and $0.78 \pm 0.08$ mg TE/mL), respectively (Figure 2). In agreement with the findings of our study, Begunova et al. [15] expressed that milk fermented by *L. reuteri* LR1 had a higher IC$_{50}$ value compared to other LABs such as *L. rhamnosus* F and *L. helveticus* NK1 on the first day of fermentation. They reported a decrease in IC$_{50}$ after 3 days of fermentation. Pihlanto et al. [30] also reported that with an increasing degree of hydrolysis, which indicates higher proteolytic activity, the rate of ACE inhibition increased. Similarly, Nielsen et al. [28] in a study on fermented milk with single cultures of *L. lactis*, *L. helveticus*, *L. acidophilus*, and *S. thermophilus* reported that the peptides obtained from *L. lactis* included higher ACE

inhibitory activity than the other three isolates studied at days 0 and 7. Tripeptides IPP and VPP are well known as the most potent ACE inhibitors in fermented milks [31]. Therefore, such peptides may be released from β-casein in the samples produced in the current study. Research conducted by Rodríguez-Gómez et al. [32] on fermented milk with *L. lactis* subsp. *lactis* also showed that this strain was able to produce hydrolyzed proteins with ACE inhibitory activity and decrease systolic and diastolic blood pressure. Similar to many previous studies, in the current study, a direct relationship was found between proteolytic activity and antihypertensive potential (T2, T3, T4, and T7).

**Table 1.** ACE inhibitory activity of crude peptide extracts obtained from fermented milk samples at three concentrations after fermentation (day 0) and after 7 days of storage.

| Treatment | Time (Day) | Concentration (mg/mL) | | |
| --- | --- | --- | --- | --- |
| | | **20** | **40** | **80** |
| T1 | 0 | $40.65 \pm 3.72$ [t] | $63.90 \pm 1.06$ [no] | $77.15 \pm 1.77$ [ghij] |
| | 7 | $37.19 \pm 1.42$ [u] | $66.1 \pm 1.24$ [lmn] | $79.93 \pm 0.65$ [efg] |
| T2 | 0 | $13.19 \pm 2.56$ [x] | $47.83 \pm 0.99$ [r] | $64.37 \pm 0.52$ [no] |
| | 7 | $9.71 \pm 0.89$ [x] | $62.93 \pm 0.60$ [no] | $70.41 \pm 1.27$ [k] |
| T3 | 0 | $61.22 \pm 0.71$ [op] | $78.41 \pm 0.30$ [fghi] | $87.01 \pm 2.40$ [b] |
| | 7 | $49.02 \pm 0.79$ [r] | $58.14 \pm 1.12$ [pq] | $82.94 \pm 2.27$ [cde] |
| T4 | 0 | $47.97 \pm 1.89$ [r] | $64.64 \pm 0.75$ [no] | $77.65 \pm 1.37$ [fghij] |
| | 7 | $31.50 \pm 4.16$ [v] | $75.13 \pm 0.61$ [ij] | $84.15 \pm 2.84$ [bcd] |
| T5 | 0 | $43.01 \pm 1.76$ [st] | $69.10 \pm 0.70$ [kl] | $81.06 \pm 0.51$ [def] |
| | 7 | $45.67 \pm 1.56$ [rs] | $68.31 \pm 1.29$ [klm] | $82.61 \pm 1.70$ [cde] |
| T6 | 0 | $55.53 \pm 1.59$ [q] | $70.17 \pm 1.41$ [k] | $84.80 \pm 3.57$ [bc] |
| | 7 | $48.29 \pm 1.68$ [r] | $76.11 \pm 0.63$ [hij] | $76.44 \pm 2.02$ [ghij] |
| T7 | 0 | $47.66 \pm 4.61$ [r] | $74.24 \pm 0.83$ [j] | $92.68 \pm 3.60$ [a] |
| | 7 | $56.95 \pm 0.86$ [q] | $65.53 \pm 1.55$ [mn] | $79.67 \pm 0.71$ [efgh] |
| T8 | 0 | $11.87 \pm 2.74$ [x] | $24.05 \pm 3.10$ [w] | $48.11 \pm 1.18$ [r] |
| | 7 | $11.87 \pm 2.74$ [x] | $24.05 \pm 3.10$ [w] | $48.11 \pm 1.18$ [r] |

Treatments included: fermented milk with *L. delbrueckii* subsp. *bulgaricus* ORT2 (T1), *L. reuteri* SRM2 (T2), *L. lactis* subsp. *lactis* BRM3 (T3), co-cultures of *L. delbrueckii* subsp. *bulgaricus* ORT2 and *L. reuteri* SRM2 (T4), co-cultures of *L. delbrueckii* subsp. *bulgaricus* ORT2 and *L. lactis* subsp. *lactis* BRM3 (T5), co-cultures of *L. reuteri* SRM2 and *L. lactis* subsp. *lactis* BRM3 (T6), and co-cultures of *L. delbrueckii* subsp. *bulgaricus* ORT2, *L. reuteri* SRM2 and *L. lactis* subsp. *lactis* BRM3 (T7). Acidified milk with lactic acid (T8). Different lowercase letters (superscripts) next to the means indicate a significant difference ($p < 0.05$) in Duncan's test. Data are presented as mean $\pm$ standard deviation of three replications.

Among the samples of milk fermented by co-cultures, sample T7 had the lowest $IC_{50}$ value ($0.61 \pm 0.08$ mg/mL) at the end of fermentation. After 7 days of cold storage, the lowest $IC_{50}$ values were observed for samples T6 and T7. The results of this study showed that peptide extracts obtained from co-cultures had higher ACE inhibitory activity than samples obtained from single cultures except T3. The highest and lowest $IC_{50}$ values were related to samples T2 and T7, respectively. During storage, ACE inhibitory activity in samples T1 and T5 decreased significantly ($p < 0.05$), while peptide content increased ($p < 0.05$, Figure 1). As concluded previously, the progress in proteolysis to a certain limit improves the activity of ACE inhibitory peptides, but excessive proteolysis may result in decreased activity [33,34]. Indeed, peptides with ACE inhibitory activity may be broken down into amino acids or smaller peptides with lower ACE inhibitory activity; which can be due to the different specificity of bacterial peptidases [28,35]. Similar to our results, Şanli et al. [17] also reported that in Kefir containing *Lb. acidophilus*, *S. thermophilus*, and *B. animalis* subsp. *lactis*, a decrease in ACE inhibitory activity was observed from day 1 of storage (92.23%) to day 28 of storage (44.25%).

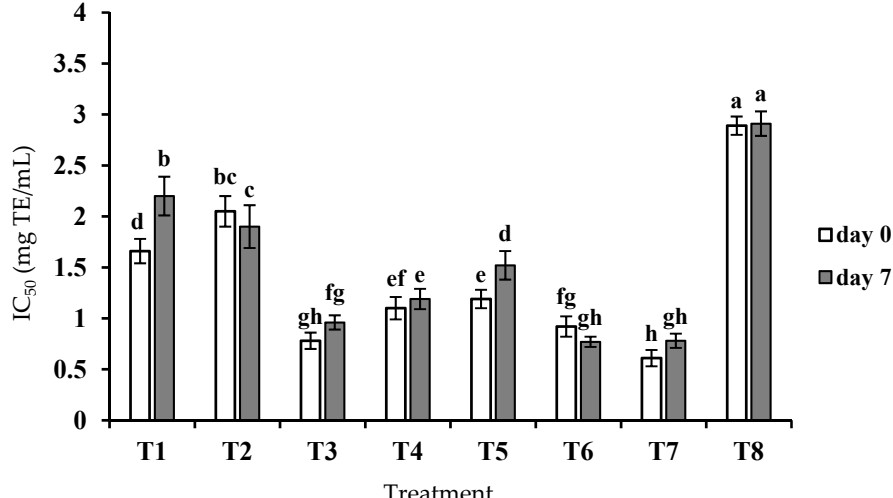

**Figure 2.** ACE inhibitory activity based on IC$_{50}$ of peptide content equivalent to tryptone in fermented milk samples after fermentation (day 0) and after 7 days of storage. T1, T2, and T3 indicate fermented milk with *L. delbruekii* subsp. *bulgaricus* ORT2, *L. reuteri* SRM2, and *L. lactis* subsp. *lactis* BRM3, respectively; T4: fermented milk with *L. delbruekii* subsp. *bulgaricus* ORT2 and *L. reuteri* SRM2; T5: fermented milk with *L. delbruekii* subsp. *bulgaricus* ORT2 and *L. lactis* subsp. *lactis* BRM3; T6: fermented milk with *L. lactis* subsp. *lactis* BRM3 and *L. reuteri* SRM2; T7: fermented milk with *L. delbruekii* subsp. *bulgaricus* ORT2, *L. reuteri* SRM2 and *L. lactis* subsp. *lactis* BRM3; T8: acidified milk with lactic acid. Different lowercase letters on each column show significant differences ($p < 0.05$) among the fermented samples during 7 days of storage.

### 3.3. Antioxidant Activity

### 3.3.1. DPPH Radical Inhibitory Activity

The results of DPPH radical inhibitory activity are presented in Table 2. There were significant differences ($p < 0.05$) in the DPPH radical inhibition activity of the control compared to samples fermented by LAB. Moreover, increasing the concentration and storage time showed a significant increase ($p < 0.05$) in DPPH radical inhibition. However, no significant differences in DPPH radical inhibition were observed between the samples fermented by single cultures and those fermented by co-cultures at a concentration of 80 mg/mL after 7 days of storage. Similar to our findings, Shori et al. [36] also reported that the DPPH radical inhibitory activity of fermented milk increased during storage for 28 days. In the study of Qian et al. [37], peptides produced in milk fermented by *L. delbrueckii* subsp. *bulgaricus* exhibited DPPH inhibitory activity.

**Table 2.** Antioxidant activity as a percentage of DPPH radical inhibition by peptide extracts from fermented milk samples obtained just after fermentation (day 0) and after 7 days of storage.

| Treatment | Time (Day) | Concentration (mg/mL) | | |
|:---:|:---:|:---:|:---:|:---:|
| | | **20** | **40** | **80** |
| T1 | 0 | 43.80 ± 2.78 [u] | 75.57 ± 0.77 [n] | 91.50 ± 1.03 [cde] |
| | 7 | 45.33 ± 1.85 [u] | 80.70 ± 1.20 [lm] | 94.43 ± 0.68 [ab] |
| T2 | 0 | 31.70 ± 1.84 [v] | 75.87 ± 0.07 [n] | 93.50 ± 1.03 [abc] |
| | 7 | 51.50 ± 0.25 [rs] | 79.40 ± 1.54 [m] | 95.47 ± 0.88 [a] |
| T3 | 0 | 52.20 ± 0.70 [qr] | 84.70 ± 0.70 [hij] | 91.40 ± 0.72 [cde] |
| | 7 | 60.97 ± 1.05 [o] | 87.60 ± 0.73 [fg] | 93.53 ± 1.11 [ab] |
| T4 | 0 | 47.80 ± 3.48 [t] | 80.43 ± 0.82 [lm] | 89.27 ± 1.86 [ef] |
| | 7 | 50.60 ± 1.20 [rs] | 85.50 ± 1.20 [ghi] | 93.60 ± 1.35 [abc] |
| T5 | 0 | 44.60 ± 2.09 [u] | 79.57 ± 2.05 [m] | 90.30 ± 0.00 [de] |
| | 7 | 56.10 ± 2.79 [p] | 81.23 ± 1.96 [klm] | 93.40 ± 0.00 [abc] |

**Table 2.** *Cont.*

| Treatment | Time (Day) | Concentration (mg/mL) | | |
|:---:|:---:|:---:|:---:|:---:|
| | | **20** | **40** | **80** |
| T6 | 0 | 47.80 ± 0.70 ᵗ | 81.50 ± 0.70 ᵏˡᵐ | 92.53 ± 1.10ᵇᶜᵈ |
| | 7 | 49.60 ± 0.46 ˢᵗ | 86.73 ± 1.20 ᵍʰⁱ | 93.37 ± 0.62 ᵃᵇᶜ |
| T7 | 0 | 47.80 ± 1.84 ᵗ | 82.47 ± 0.91 ʲᵏˡ | 90.30 ± 0.00 ᵈᵉ |
| | 7 | 53.90 ± 0.56 �q | 83.43 ± 1.95 ⁱʲᵏ | 94.43 ± 0.68 ᵃᵇ |
| T8 | 0 | 13.30 ± 1.3 ʸ | 13.50 ± 0.70 ˣʸ | 15.60 ± 0.6 ʷˣ |
| | 7 | 12.40 ± 0.4 ʸ | 16.25 ± 1.25 ʷ | 16.20 ± 1.2 ʷ |

Treatments included: fermented milk with *L. delbrueckii* subsp. *bulgaricus* ORT2 (T1), *L. reuteri* SRM2 (T2), *L. lactis* subsp. *lactis* BRM3 (T3), co-cultures of *L. delbrueckii* subsp. *bulgaricus* ORT2 and *L. reuteri* SRM2 (T4), co-cultures of *L. delbrueckii* subsp. *bulgaricus* ORT2 and *L. lactis* subsp. *lactis* BRM3 (T5), co-cultures of *L. reuteri* SRM2 and *L. lactis* subsp. *lactis* BRM3 (T6), and co-cultures of *L. delbrueckii* subsp. *bulgaricus* ORT2, *L. reuteri* SRM2 and *L. lactis* subsp. *lactis* BRM3 (T7). Acidified milk with lactic acid (T8). Different lowercase letters (superscripts) next to the means indicate a significant difference ($p < 0.05$) in Duncan's test. Data are presented as mean ± standard deviation of three replications.

In our study, although *L. reuteri* SRM2 showed lower proteolytic activity than *L. delbrueckii* ORT2 and *L. lactis* subsp. *lactis* BRM3, but resulted in comparable DPPH scavenging activity. In this regard, Ayyash et al. [13] also reported that although *L. reuteri* showed lower proteolytic activity than *L. plantarum*, the milk fermented by *L. reuteri* showed higher DPPH inhibitory activity. Differences in antioxidant activity may be due to differences in LAB strains and their type of proteolytic enzymes [38], and therefore not only the quantity but also the nature of the produced peptides and amino acids.

### 3.3.2. Ferric-Reducing Antioxidant Power (FRAP)

Significant differences ($p < 0.05$) were observed in the FRAP of the control and fermented samples with LAB (Table 3). As the concentration of the peptide extracts increased, the FRAP activity of all samples increased. After 7 days of storage, a significant increase ($p < 0.05$) was observed in the FRAP of samples T1, T3, T6, and T7 at a concentration of 80 mg/mL. At this concentration and time, no significant difference was found among the samples fermented by single cultures and those fermented by co-cultures in terms of FRAP activity.

**Table 3.** Antioxidant activity as Ferric-Reducing Antioxidant Power (FRAP *) of peptide extracts from fermented milk samples after fermentation (day 0) and after 7 days of storage.

| Treatment | Time (Day) | Concentration (mg/mL) | | |
|:---:|:---:|:---:|:---:|:---:|
| | | **20** | **40** | **80** |
| T1 | 0 | 0.29 ± 0.01ᴾ | 0.41 ± 0.01 ˡᵐⁿᵒ | 0.57 ± 0.01 ᵍʰⁱʲ |
| | 7 | 0.53 ± 0.00 ʰⁱʲᵏ | 0.62 ± 0.19 ᵍʰ | 0.86 ± 0.04 ᵇᶜᵈ |
| T2 | 0 | 0.39 ± 0.01 ˡᵐⁿᵒ | 0.54 ± 0.01 ʰⁱʲᵏ | 0.89 ± 0.01 ᵃᵇ |
| | 7 | 0.38 ± 0.00 ᵐⁿᵒᵖ | 0.45 ± 0.12 ᵏˡᵐ | 0.86 ± 0.01 ᵃᵇᶜᵈ |
| T3 | 0 | 0.31 ± 0.01 ᵒᵖ | 0.42 ± 0.01 ˡᵐⁿ | 0.77 ± 0.01 ᵈᵉ |
| | 7 | 0.37 ± 0.02 ᵐⁿᵒᵖ | 0.56 ± 0.04 ᵍʰⁱʲ | 0.89 ± 0.01 ᵃᵇ |
| T4 | 0 | 0.40 ± 0.00 ˡᵐⁿᵒ | 0.71 ± 0.11 ᵉᶠ | 0.96 ± 0.05 ᵃ |
| | 7 | 0.57 ± 0.02 ᵍʰⁱʲ | 0.86 ± 0.02 ᵃᵇᶜᵈ | 0.96 ± 0.01 ᵃ |
| T5 | 0 | 0.57 ± 0.02 ᵍʰⁱʲ | 0.80 ± 0.02 ᵇᶜᵈᵉ | 0.95 ± 0.05 ᵃ |
| | 7 | 0.59 ± 0.01 ᵍʰⁱ | 0.92 ± 0.02 ᵃ | 0.94 ± 0.01 ᵃ |
| T6 | 0 | 0.37 ± 0.03 ᵐⁿᵒᵖ | 0.48 ± 0.01 ʲᵏˡ | 0.65 ± 0.04 ᶠᵍ |
| | 7 | 0.34 ± 0.02 ⁿᵒᵖ | 0.52 ± 0.22 ⁱʲᵏ | 0.88 ± 0.01 ᵃᵇᶜ |
| T7 | 0 | 0.34 ± 0.02 ⁿᵒᵖ | 0.45 ± 0.02 ᵏˡᵐ | 0.79 ± 0.01 ᶜᵈᵉ |
| | 7 | 0.29 ± 0.01 ᴾ | 0.39 ± 0.02 ˡᵐⁿᵒ | 0.89 ± 0.01 ᵃᵇ |

**Table 3.** *Cont.*

| Treatment | Time (Day) | Concentration (mg/mL) | | |
| | | 20 | 40 | 80 |
| --- | --- | --- | --- | --- |
| T8 | 0 | 0.08 ± 0.01 [q] | 0.12 ± 0.01 [q] | 0.16 ± 0.01 [q] |
| | 7 | 0.07 ± 0.02 [q] | 0.15 ± 0.01 [q] | 0.13 ± 0.02 [q] |

* Expressed as the absorbance intensity at 760 nm. Treatments included: fermented milk with *L. delbrueckii* subsp. *bulgaricus* ORT2 (T1), *L. reuteri* SRM2 (T2), *L. lactis* subsp. *lactis* BRM3 (T3), co-cultures of *L. delbrueckii* subsp. *bulgaricus* ORT2 and *L. reuteri* SRM2 (T4), co-cultures of *L. delbrueckii* subsp. *bulgaricus* ORT2 and *L. lactis* subsp. *lactis* BRM3 (T5), co-cultures of *L. reuteri* SRM2 and *L. lactis* subsp. *lactis* BRM3 (T6), and co-cultures of *L. delbrueckii* subsp. *bulgaricus* ORT2, *L. reuteri* SRM2 and *L. lactis* subsp. *lactis* BRM3 (T7). Acidified milk with lactic acid (T8). Different lowercase letters (superscripts) next to the means indicate a significant difference ($p < 0.05$) in Duncan's test. Data are presented as mean ± standard deviation of three replications.

In the study on the antioxidant properties of casein hydrolysates from ovine milk, Corrêa et al. [39] concluded that a compound that has a high ability to inhibit DPPH radicals may not necessarily have a high FRAP, as the mechanisms of the reactions are different. Such a finding was also observed in the present study, as the peptide extract from sample T2 showed the highest DPPH scavenging activity, while samples T4 and T5 showed the highest FRAP activity on day 0. Correa et al. [39] also found that with increasing proteolytic activity, the reducing power increased. Despite the general positive correlation between the reducing power and the proteolytic activity, the reducing power decreased with a further increase in proteolytic activity. In another work, storage time caused a significant effect on increasing the FRAP in milk fermented by LAB species [40,41].

### 3.3.3. Hydroxyl Radical Scavenging Activity

OH-scavenging activity of the fermented samples followed a time- and concentration-dependent pattern ($p < 0.05$) (Table 4). The highest activity was observed for samples T1, T2, and T7 (95.37, 95.43, and 96.37%, respectively), the latter of which was fermented by triple co-cultures. Unlike samples T1 and T2, whose increase in its OH-scavenging activity during storage is consistent with their increase in proteolytic activity, sample T7 also exhibited a significant increase in its OH-scavenging activity during storage, despite a decrease in proteolysis. Thus, the OH-scavenging activity of peptide extracts was not directly related to the proteolytic activity of the strains. Accordingly, it can be concluded that the interaction between cultured bacterial strains can result in the production of other bioactive compounds with antioxidant properties during fermentation, in addition to bioactive peptides. Consistently with our results, Tyagi et al. [42] reported that *L. reuteri* showed higher ability to release antioxidant compounds such as phenolic compounds in fermented brown rice, in comparison to other LABs. Similar to the current findings, the OH-scavenging activity of goat milk fermented by *L. fermentum* [43] and *L. casei* L61 [44] was increased with increasing storage time.

### 3.3.4. Total Antioxidant Activity

Significant differences were found in the total antioxidant activity of the samples expressed as their capacity to reduce $Mo^{+6}$ to $Mo^{+5}$ (Table 5). Increased concentration of peptide extracts enhanced the total antioxidant activity of all fermented samples ($p < 0.05$), but the effect of storage for 7 days varied. According to this assay, the co-culture of *L. reuteri* SRM2 and *L. lactis* subsp. *lactis* BRM3 exhibited the best bacterial interaction for the production of antioxidant peptides. The antioxidative properties of the peptides obtained from food proteins are associated with the structure, composition, and hydrophobicity of the peptides [45]. Other researchers stated that the total antioxidant activity of fermented milk samples increased ($p < 0.05$) until day 6 and then decreased significantly until day 21 [16]. This might be due to the breakdown of antioxidant peptides by proteolytic enzymes during storage time.

**Table 4.** Antioxidant activity as OH-scavenging (%) in peptide extracts from fermented milk samples after fermentation (day 0) and after 7 days of storage.

| Treatment | Time (Day) | Concentration (mg/mL) | | |
|---|---|---|---|---|
| | | 20 | 40 | 80 |
| T1 | 0 | 1.00 ± 0.87 [tu] | 46.50 ± 1.37 [h] | 90.03 ± 2.00 [d] |
| | 7 | 3.90 ± 0.2 [rs] | 51.37 ± 2.49 [g] | 95.37 ± 0.94 [ab] |
| T2 | 0 | 0.00 ± 0.00 [u] | 30.80 ± 1.78 [k] | 78.53 ± 3.37 [f] |
| | 7 | 6.20 ± 1.36 [qr] | 39.37 ± 1.29 [i] | 95.43 ± 2.72 [ab] |
| T3 | 0 | 2.67 ± 0.47 [st] | 27.10 ± 2.14 [l] | 92.30 ± 1.59 [c] |
| | 7 | 3.97 ± 0.58 [rs] | 39.33 ± 1.08 [i] | 93.43 ± 0.36 [bc] |
| T4 | 0 | 2.30 ± 0.00 [stu] | 31.20 ± 1.21 [k] | 88.30 ± 1.03 [de] |
| | 7 | 6.57 ± 1.08 [q] | 37.33 ± 0.10 [ij] | 93.57 ± 1.95 [bc] |
| T5 | 0 | 1.60 ± 0.82 [stu] | 35.43 ± 1.95 [j] | 88.77 ± 1.53 [de] |
| | 7 | 0.97 ± 0.88 [tu] | 21.83 ± 1.34 [mn] | 86.50 ± 1.03 [e] |
| T6 | 0 | 0.00 ± 0.00 [u] | 21.10 ± 0.88 [n] | 87.70 ± 0.00 [de] |
| | 7 | 0.67 ± 0.31 [tu] | 23.50 ± 0.18 [m] | 88.37 ± 0.98 [de] |
| T7 | 0 | 1.60 ± 0.94 [qr] | 44.73 ± 3.12 [h] | 93.23 ± 1.70 [bc] |
| | 7 | 7.40 ± 1.26 [pq] | 50.83 ± 1.96 [g] | 96.37 ± 0.10 [a] |
| T8 | 0 | 0.00 ± 0.00 [u] | 8.40 ± 0.10 [pq] | 11.40 ± 0.10 [o] |
| | 7 | 0.60 ± 0.10 [tu] | 9.75 ± 0.10 [op] | 11.75 ± 0.10 [o] |

Treatments included: fermented milk with *L. delbrueckii* subsp. *bulgaricus* ORT2 (T1), *L. reuteri* SRM2 (T2), *L. lactis* subsp. *lactis* BRM3 (T3), co-cultures of *L. delbrueckii* subsp. *bulgaricus* ORT2 and *L. reuteri* SRM2 (T4), co-cultures of *L. delbrueckii* subsp. *bulgaricus* ORT2 and *L. lactis* subsp. *lactis* BRM3 (T5), co-cultures of *L. reuteri* SRM2 and *L. lactis* subsp. *lactis* BRM3 (T6), and co-cultures of *L. delbrueckii* subsp. *bulgaricus* ORT2, *L. reuteri* SRM2 and *L. lactis* subsp. *lactis* BRM3 (T7). Acidified milk with lactic acid (T8). Different lowercase letters (superscripts) next to the means indicate a significant difference ($p < 0.05$) in Duncan's test. Data are presented as mean ± standard deviation of three replications.

**Table 5.** Total antioxidant activity of crude peptide extracts from fermented milk samples obtained just after fermentation (day 0) and after 7 days of storage.

| Treatment | Time (Day) | Concentration (mg/mL) | | |
|---|---|---|---|---|
| | | 20 | 40 | 80 |
| T1 | 0 | 0.35 ± 0.01 [q] | 0.66 ± 0.01 [gh] | 0.86 ± 0.01 [d] |
| | 7 | 0.26 ± 0.01 [s] | 0.58 ± 0.01 [k] | 0.85 ± 0.02 [d] |
| T2 | 0 | 0.37 ± 0.01 [pq] | 0.61 ± 0.03 [j] | 0.90 ± 0.01 [c] |
| | 7 | 0.42 ± 0.03 [o] | 0.63 ± 0.03 [ij] | 0.89 ± 0.02 [c] |
| T3 | 0 | 0.42 ± 0.01 [o] | 0.65 ± 0.01 [hi] | 0.76 ± 0.02 [f] |
| | 7 | 0.36 ± 0.00 [pq] | 0.45 ± 0.01 [n] | 0.68 ± 0.03 [g] |
| T4 | 0 | 0.37 ± 0.03 [pq] | 0.61 ± 0.02 [j] | 0.96 ± 0.02 [b] |
| | 7 | 0.32 ± 0.01 [r] | 0.41 ± 0.02 [o] | 0.89 ± 0.00 [c] |
| T5 | 0 | 0.32 ± 0.02 [r] | 0.38 ± 0.01 [p] | 0.77 ± 0.01 [ef] |
| | 7 | 0.31 ± 0.01 [r] | 0.35 ± 0.01 [q] | 0.79 ± 0.00 [e] |
| T6 | 0 | 0.37 ± 0.02 [pq] | 0.43 ± 0.01 [no] | 0.67 ± 0.03 [gh] |
| | 7 | 0.37 ± 0.02 [pq] | 0.48 ± 0.00 [m] | 0.75 ± 0.01 [f] |
| T7 | 0 | 0.36 ± 0.00 [pq] | 0.56 ± 0.01 [l] | 0.99 ± 0.01 [a] |
| | 7 | 0.36 ± 0.00 [pq] | 0.45 ± 0.01 [n] | 0.68 ± 0.01 [g] |
| T8 | 0 | 0.06 ± 0.01 [vw] | 0.08 ± 0.01 [u] | 0.12 ± 0.01 [t] |
| | 7 | 0.04 ± 0.01 [w] | 0.07 ± 0.01 [uv] | 0.13 ± 0.01 [t] |

Treatments included: fermented milk with *L. delbrueckii* subsp. *bulgaricus* ORT2 (T1), *L. reuteri* SRM2 (T2), *L. lactis* subsp. *lactis* BRM3 (T3), co-cultures of *L. delbrueckii* subsp. *bulgaricus* ORT2 and *L. reuteri* SRM2 (T4), co-cultures of *L. delbrueckii* subsp. *bulgaricus* ORT2 and *L. lactis* subsp. *lactis* BRM3 (T5), co-cultures of *L. reuteri* SRM2 and *L. lactis* subsp. *lactis* BRM3 (T6), and co-cultures of *L. delbrueckii* subsp. *bulgaricus* ORT2, *L. reuteri* SRM2 and *L. lactis* subsp. *lactis* BRM3 (T7). Acidified milk with lactic acid (T8). Different lowercase letters (superscripts) next to the means indicate a significant difference ($p < 0.05$) in Duncan's test. Data are presented as mean ± standard deviation of three replications.



## 4. Conclusions

Proteolytic LAB strains used as single or co-cultures displayed remarkable potential to manufacture fermented milk with ACE inhibitory and antioxidant activities. Overall, higher proteolysis resulted in lower $IC_{50}$ and therefore higher ACE inhibition. In terms of antioxidant activity, for most samples (except for samples fermented with *L. reuteri*), a direct relationship with proteolytic activity was found; however, it is noteworthy that there was no positive correlation between the antioxidant activities of a sample for all assays necessarily. This implies that the fermented milk-derived peptides exhibit their antioxidant activity in different ways, as described before. These results support the idea of the loading of different LAB cultures to boost the functional and health-promoting attributes of fermented dairy products. It is suggested that in future research the sequence and composition of amino acids released in fermented milk with *L. delbrueckii* subsp. *bulgaricus* ORT2, *L. reuteri* SRM2 and *L. lactis* subsp. *lactis* BRM3 are determined. Moreover, for the industrial development of functional fermented milk, its proven beneficial properties including antihypertensive and antioxidant activity should be studied in vivo.

**Author Contributions:** Conceptualization, S.L. and A.M.; methodology, S.L. and A.M.; investigation, S.L. and A.M.; formal analysis, M.M.; data curation, A.M. and M.M.; writing-original draft preparation, M.M.; writing-review and editing, A.M., L.G.G.-M., F.G. and M.K.; supervision, A.M. and F.G.; project administration, A.M. All authors have read and agreed to the published version of the manuscript.

**Funding:** This research received no external funding. F.G would like to acknowledge Enterprise Ireland and the European Union's Horizon 2020 Programme under the Marie Skłodowska-Curie Career-Fit Plus Action [Grant MF20200171] as his funding agency.

**Institutional Review Board Statement:** Not applicable.

**Informed Consent Statement:** Not applicable.

**Data Availability Statement:** Not applicable.

**Acknowledgments:** The authors would like to thank Gorgan University of Agricultural Sciences and Natural Resources for the financial support of this study.

**Conflicts of Interest:** The authors declare no conflict of interest.

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
