# Peer review of "Single and Co-Cultures of Proteolytic Lactic Acid Bacteria in the Manufacture of Fermented Milk with High ACE Inhibitory and Antioxidant Activities"

_fermentation, doi:10.3390/fermentation8090448_

Round 1
Reviewer 1 Report
The topic of this manuscript is very interesting from the point of view of fermented food production and the impact of this food on human health. The topic of this manuscript fits the FERMENTATION journal, which is why I think it should be allowed to be published in this journal.
The manuscript is very well prepared. The purpose of the research has been clearly described and justified by well-chosen references. It is worth emphasizing that the Authors used the current taxonomic classification of lactic acid bacteria. This is praiseworthy.
This manuscript contains some deficiencies of information, which is why I think that this must be improved before allowing the manuscript to be published. My suggestions are as follows:
· Section „2. Materials and methods” – Methodology is very well described, I have no objections to this. However, in my opinion, information on the number of repetitions of the experiment and the number of repetitions of individual analyzes should be supplemented, it is very important for the statistical analysis of the results. In addition, when describing centrifuging parameters, the information about the temperature of the centrifuge process (lines 100, 118, and 145) should be added.
· Section „3. Results and Discussion” – The results are well presented and discussed. However, there is a lack of looking for a correlation between the results received (according to the manuscript title). It is a pity that the Authors did not mark the population of lactic acid bacteria (at least after 7 days of sample storage), this information could help with the interpretation of certain results.
· Section „4. Conclusion” – I have no conclusion in accordance with the manuscript title: how ACE-inhibitors and antioxidant activities depend on proteolytic activities of LAB?
Author Response
Reviewer 1
Section „2. Materials and methods” – Methodology is very well described, I have no objections to this. However, in my opinion, information on the number of repetitions of the experiment and the number of repetitions of individual analyzes should be supplemented, it is very important for the statistical analysis of the results. In addition, when describing centrifuging parameters, the information about the temperature of the centrifuge process (lines 100, 118, and 145) should be added.
Response: Thanks a lot for your suggestion. It was included in the text.
Section „3. Results and Discussion” – The results are well presented and discussed. However, there is a lack of looking for a correlation between the results received (according to the manuscript title). It is a pity that the Authors did not mark the population of lactic acid bacteria (at least after 7 days of sample storage), this information could help with the interpretation of certain results.
Response: To investigate cell viability in mixed cultures such as T4, T5, T6 and T7, it is necessary to count bacteria (L. delbruekii, L. reuteri and L. lactis) in specific culture media, or to use RT-PCR which was not possible at the time of investigation.
Section „4. Conclusion” – I have no conclusion in accordance with the manuscript title: how ACE-inhibitors and antioxidant activities depend on proteolytic activities of LAB?
Response: ACE inhibition and antioxidant activities of fermented milk samples were previously attributed to factors such as bioactive peptides (Ayyash et al., 2018; Kim et al., 2021; Begunova et al., 2021; Li et al., 2020). But in all the aforementioned researches, for all samples, a direct relationship between proteolytic activity and measured indicators was not found. In agreement with our findings, Ayyash et al., 2018 also reported that although L. reuteri showed lower proteolytic activity than L. plantarum, the milk fermented by L. reuteri showed higher DPPH inhibitory activity. Begunova et al. (2021) also showed that the antioxidant activity of the milk fermented by L. helveticus NK1, L. reuteri LR1 and L. rhamnosus F increased with the strains’ proteolytic activity. For the angiotensin I-converting enzyme (ACE) inhibitory activity, the same tendency was not observed. Although the proteolytic activity of L. helveticus NK1 was higher than that of L. rhamnosus F, the ACE inhibition of milk fermented by L. helveticus NK1 was lower than L. rhamnosus F. Also as commented by Rubak et al. (2022), high peptide content was not always associated with high ACEI activity in samples. ACEI activity is more related to the abundance of ACEI peptides that could be released during fermentation.
In our study, a direct relationship between proteolytic activity and antihypertensive and antioxidant activities of some treatments (T2, T3, T4 and T7) has been observed (line 219-220). Increased OH-scavenging activity in sample T1 and T2 during storage is consistent with their increase in proteolytic activity (line 304-305). After 7 days of storage, total antioxidant activity of samples T3, T4 and T7 indicated a significant decrease (P<0.05) which was consistent with the reduction of their proteolytic activity. We tried to conclude the relationship between proteolytic activity and ACE inhibitory/antioxidant activity in "CONCLUSION".
- Ayyash, M.; Al-Dhaheri, A.S.; Al Mahadin, S.; Kizhakkayil, J.; Abushelaibi, A. In vitro investigation of anticancer, antihypertensive, antidiabetic, and antioxidant activities of camel milk fermented with camel milk probiotic: A comparative study with fermented bovine milk. Dairy Sci. 2018, 101, 900-911.
- Kim, E.D.; Lee, H.S.; kim, K.T.; Paik, H.D. Antioxidant and Angiotensin-Converting Enzyme (ACE) Inhibitory Activities of Yogurt Supplemented with Lactiplantibacillus plantarum NK181 and Lactobacillus delbrueckii KU200171 and Sensory Evaluation. Foods 2021, 10, 2324.
- Begunova, A.V.; Savinova, O.S.; Glazunova, O.A.; Moiseenko, K.V.; Rozhkova, I.V.; Fedorova, T.V. Development of Antioxidant and Antihypertensive Properties during Growth of Lactobacillus helveticus, Lactobacillus rhamnosus and Lactobacillus reuteri on Cow’s Milk: Fermentation and Peptidomics Study. Foods 2021, 10, 17.
- Li, S.N.; Tang, S.H.; He, Q.; Hu, J.X.; Zheng, J. In vitro antioxidant and angiotensin-converting enzyme inhibitory activity of fermented milk with different culture combinations. Dairy Sci. 2020, 103, 1120-1130.
- Șanli, T.; Akal, H.C.; Yetisemiyen, A.; HayaLoglu, A.A. Influence of adjunct cultures on angiotensin-converting enzyme (ACE)-inhibitory activity, organic acid content and peptide profile of kefir. J. Dairy Technol., 2018, 71, 131-139.
- Rubak, Y.T.; Nuraida, L.; Iswantini, D.; Prangdimurti, E. Angiotensin-I-Converting enzyme inhibitory peptides in goat milk fermented by lactic acid bacteria isolated from fermented food and breast milk. food Sci. Anim. Resour., 2022, 42, 46-60.
Reviewer 2 Report
Thank you for the possibility of reviewing this interesting paper. The study links two current topics of functional food: ACE - inhibitors with fermented milk products. However, the study may be corrected and the research model enriched to result in higher scientific soundness.
Here are my detailed remarks:
Line 90. The better would be if this part sounded as follows: "...or 1:1 combination of isolates..."
Line 203. You refer to data that are not presented in the work. Thus, your statements are less reliable. Please at least provide some summary of this data, e.g. as an Appendix or Supplementary material.
Line 215. Please analyze the literature on the subject and check if somebody examined the structure of the fermented milk proteins. Also please discuss the pharmacophore structure of ACE inhibitors and use this information to predict the structure of proteins in your samples.
Lines 331-2. And how this information corresponds with the findings of your study?
The last but not least is my general remark. It's good that you analyzed the proteolysis and peptide content but this paper very lacks information on the molecular mass of resulted proteins. It's a pity that you did not perform electrophoresis of proteins to check their Dalton mass. Such information would be stronger evidence that different LAB differ in protein hydrolysis as well as would stronger correspond with ACE study.
Author Response
Reviewer 2
Line 90. The better would be if this part sounded as follows: "...or 1:1 combination of isolates..."
Response: It was included in the text.
Line 203. You refer to data that are not presented in the work. Thus, your statements are less reliable. Please at least provide some summary of this data, e.g. as an Appendix or Supplementary material.
Response: Many thanks for your helpful comment, the results are presented in the text (Table 1).
Line 215. Please analyze the literature on the subject and check if somebody examined the structure of the fermented milk proteins. Also please discuss the pharmacophore structure of ACE inhibitors and use this information to predict the structure of proteins in your samples.
Response: By examining the pharmacophore models for ACE inhibitors, it is suggested that a tri-peptide (thr-val-phe) may determine the structure of ACE inhibitory activity (Wei et al., 2008). In addition, tripeptides IPP and VPP are well known as potent ACE inhibitors in fermented milks (Nakamura et al., 1995). The discussion was improved in the text.
Wei, W.; ShengRong, S.; FengQin, f.; GuoQing, H.; ZhanLi, W. Pharmacophore-based structure optimization of angiotensin converting enzyme inhibitory peptide. Sci. China, Ser. B-Chem., 2008, 51, 786-793.
Nakamura, Y.; Yamamoto, N.; Sakai, K.; Okubo, A.; Yamazaki, S.; Takano, T. Purification and characterization of angiotensin I-converting enzyme inhibitors from sour milk. J Dairy Sci. 1995, 78, 777–783.
Lines 331-2. And how this information corresponds with the findings of your study?
Line 331-332: This was a general description of the properties of antioxidant peptides from food proteins. The aim was to show a relationship between the type of peptides and antioxidant properties.
The last but not least is my general remark. It's good that you analyzed the proteolysis and peptide content but this paper very lacks information on the molecular mass of resulted proteins. It's a pity that you did not perform electrophoresis of proteins to check their Dalton mass. Such information would be stronger evidence that different LAB differ in protein hydrolysis as well as would stronger correspond with ACE study.
Response: Yes, I agree. In order to show the difference in proteolytic activity of LABs, electrophoresis could show a better understanding than OPA alone. For the next phase, we will focus on peptides molecular weight by running electrophoresis and size exclusion chromatography.
Reviewer 3 Report
The paper entitled ‘’ Single and co-cultures of proteolytic lactic acid bacteria in the manufacture of fermented milk with high ACE inhibitory and antioxidant activities’’ treats an interesting topic related to manufacture of fermented milk with antihypertensive and antioxidant properties by using co-cultures of Lactobacillus sp., Limosilactobacillus sp. and Lactococcus sp.
There are some adjustments that need to be made:
Point 1 (Introduction):
1.1. A brief reference to the metabolism of proteolytic lactic acid bacteria strains used in this study could be useful in order to sustain the research design and the Conclusions’ part.
Point 2 (Materials and Methods):
2.1. Please indicate the type and producer of the thermal processing unit for skim milk sterilization.
2.2. Please indicate the volume of the sterile milk inoculated and incubated, type of glasses used and the model of incubator used.
2.3. Please indicate the method of enumeration of cell population (culture media used, condition of incubation).
2.4. Lines 88 – 90: ‘’ …. sterile milk was inoculated with 2% (v/v) of the suspension …. of each isolate or combination (each at 1%) of isolates (108 CFU/mL)’’. Please indicate the amount of bacteria inoculated for the sample containing all the three bacteria species (T7).
2.5. Please indicate how different the time to reach pH 4.6 for all the experimental variants was.
2.6. Please indicate the model and producer of centrifuge used to prepare crude peptide extracts.
2.7. Please indicate the model and producer of the spectrophotometer(s) used to measure the absorbance for determination of ACE inhibitory activity and antioxidant activities respectively.
2.8. Please indicate how it was expressed the Total Antioxidant Capacity.
Point 3 (Results and Discussion):
3.1. Even though the dynamics of bacteria was not determined, please indicate if factors such as cell viability during 7 days of refrigeration, bacterial symbiosis (in T4, T5, T6 and T7 respectively), and bacteria lifestyle / strains’ requirements for growth could be responsible for correlation between the analyzed parameters and / or their trends (i.e. decreasing levels of peptides and amino acids after 7 days of storage, production of bioactive compounds with antioxidant properties during fermentation, total antioxidant activity of crude peptide extracts).
Point 4 (Conclusion):
4.1. Conclusion part could be more punctual (in relationship with the studied lactic acid bacteria strains) and could underline the further area of research.
Author Response
Reviewer 3
Point 1 (Introduction):
1.1. A brief reference to the metabolism of proteolytic lactic acid bacteria strains used in this study could be useful in order to sustain the research design and the Conclusions’ part.
Response: proteolytic LAB degrades proteins into peptides by cell envelope proteinase (CEP). In the next step, the peptides are transferred into the cell and they are broken down into amino acids by peptidases (Wang et al., 2021). Proteolytic LAB strains during fermentation of dairy products reduces allergenicity by breaking down casein (Iwamoto et al., 2019). In addition, they can improve the digestibility of protein in fermented dairy products by their protease system (Wang et al., 2021).
Iwamoto, H.; Matsubara, T.; Okamoto, T.; Matsumoto, T.; Yoshikawa, M.; Takeda, Y. Ingestion of casein hydrolysate induces oral tolerance and suppresses subsequent epicutaneous sensitization and development of anaphylaxis reaction to casein in mice. Int. Arch. Allergy Imm. 2019, 179, 221–230.
Wang, Y.; Wu, J.; Lv, M.; Shao, Z.; Hungwe, M.; Wang, J.; Bai1, X.; Xie, J.; Wang, Y.; Geng, W. Metabolism Characteristics of Lactic Acid Bacteria and the Expanding Applications in Food Industry. Front. bioeng. Biotechnol. 2021, 9, Article 612285.
Point 2 (Materials and Methods):
2.1. Please indicate the type and producer of the thermal processing unit for skim milk sterilization.
Response: Skim milk was provided from Golestan Pegah factory, Gorgan, Iran. 12 g skim milk was added to distilled water, adjusted to 100 mL, and autoclaved in sealed containers at 115 °C for 15 min.
2.2. Please indicate the volume of the sterile milk inoculated and incubated, type of glasses used and the model of incubator used.
Response: 100 mL of 12% w/v reconstituted skim milk in sterile bottle was inoculated and incubated (BD115, Binder, Germany) at 37 °C.
2.3. Please indicate the method of enumeration of cell population (culture media used, condition of incubation).
Response: Counting of the cell population was done by pour plating 1 ml of the desired dilution on MRS agar and incubation at 37 °C.
2.4. Lines 88 – 90: ‘’ …. sterile milk was inoculated with 2% (v/v) of the suspension …. of each isolate or combination (each at 1%) of isolates (108 CFU/mL)’’. Please indicate the amount of bacteria inoculated for the sample containing all the three bacteria species (T7).
Response: Each bacterium was inoculated to milk in the amount of 670 μl, which in total, like other treatments, the volume of bacterial inoculation was 2 ml per 100 ml of milk.
2.5. Please indicate how different the time to reach pH 4.6 for all the experimental variants was.
Response: The time to reach pH to 4.6 varied from 8 h (sample fermented by L. delbreuckii subsp. bulgaricus and L. lactis) to 18 h (sample fermented by single culture of L. reuteri).
2.6. Please indicate the model and producer of centrifuge used to prepare crude peptide extracts.
Response: (model Combi 514R, Hanil Science Industrial, Gimpo-si, South Korea). It was included in the text.
2.7. Please indicate the model and producer of the spectrophotometer(s) used to measure the absorbance for determination of ACE inhibitory activity and antioxidant activities respectively.
Response: model T80, PG Instruments Ltd, Wibtoft, UK. It was included in the text.
2.8. Please indicate how it was expressed the Total Antioxidant Capacity.
Response: Sometimes, the molybdate reducing activity is expressed as total antioxidant activity.
Meshginfar, N.; Sadeghi Mahoonak, A.; Hosseinian, F.; Ghorbani, M.; Tsopmo, A. Production of antioxidant peptide fractions from a by-product of tomato processing: Mass spectrometry identification of peptides and stability to gastrointestinal digestion. JFST 2018, 55, 3498-3507.
Point 3 (Results and Discussion):
3.1. Even though the dynamics of bacteria was not determined, please indicate if factors such as cell viability during 7 days of refrigeration, bacterial symbiosis (in T4, T5, T6 and T7 respectively), and bacteria lifestyle / strains’ requirements for growth could be responsible for correlation between the analyzed parameters and / or their trends (i.e. decreasing levels of peptides and amino acids after 7 days of storage, production of bioactive compounds with antioxidant properties during fermentation, total antioxidant activity of crude peptide extracts).
Response: To investigate bacterial symbiosis in mixed cultures such as T4, T5, T6 and T7, it is necessary to count bacteria in specific culture media. It means to prepare a specific culture medium for each strain (L. delbruekii, L. reuteri and L. lactis), which was difficult for the authors.
Point 4 (Conclusion):
4.1. Conclusion part could be more punctual (in relationship with the studied lactic acid bacteria strains) and could underline the further area of research.
Response: It is suggested that in future research, the sequence and composition of amino acids released in fermented milk with L. delbrueckii subsp. bulgaricus ORT2, L. reuteri SRM2 and L. lactis subsp. lactis BRM3 are determined. Also, for the industrial development of functional fermented milk, its proven beneficial properties including antihypertensive and antioxidant activity should be studied in vivo.